# Mutation Analysis of Pancreatic Juice and Plasma for the Detection of Pancreatic Cancer

**DOI:** 10.3390/ijms241713116

**Published:** 2023-08-23

**Authors:** Iris J. M. Levink, Maurice P. H. M. Jansen, Zakia Azmani, Wilfred van IJcken, Ronald van Marion, Maikel P. Peppelenbosch, Djuna L. Cahen, Gwenny M. Fuhler, Marco J. Bruno

**Affiliations:** 1Department of Gastroenterology & Hepatology, Erasmus MC, University Medical Center, 3015 GD Rotterdam, The Netherlands; m.peppelenbosch@erasmusmc.nl (M.P.P.); g.fuhler@erasmusmc.nl (G.M.F.); m.bruno@erasmusmc.nl (M.J.B.); 2Department of Medical Oncology, Erasmus MC, University Medical Center, 3015 GD Rotterdam, The Netherlands; 3Center for Biomics, Erasmus MC, University Medical Center, 3015 GD Rotterdam, The Netherlandsw.vanijcken@erasmusmc.nl (W.v.I.); 4Department of Pathology, Erasmus MC, University Medical Center, 3015 GD Rotterdam, The Netherlands; r.vanmarion@erasmusmc.nl

**Keywords:** pancreatic cancer, pancreatic juice, plasma, biomarkers, DNA, mutation, detection, diagnosis, KRAS, TP53, SMAD4, cell-free DNA (cfDNA), circulating tumor DNA (ctDNA), liquid biopsy, precision medicine

## Abstract

Molecular profiling may enable earlier detection of pancreatic cancer (PC) in high-risk individuals undergoing surveillance and allow for personalization of treatment. We hypothesized that the detection rate of DNA mutations is higher in pancreatic juice (PJ) than in plasma due to its closer contact with the pancreatic ductal system, from which pancreatic cancer cells originate, and higher overall cell-free DNA (cfDNA) concentrations. In this study, we included patients with pathology-proven PC or intraductal papillary mucinous neoplasm (IPMN) with high-grade dysplasia (HGD) from two prospective clinical trials (KRASPanc and PACYFIC) for whom both PJ and plasma were available. We performed next-generation sequencing on PJ, plasma, and tissue samples and described the presence (and concordance) of mutations in these biomaterials. This study included 26 patients (25 PC and 1 IPMN with HGD), of which 7 were women (27%), with a median age of 71 years (IQR 12) and a median BMI of 23 kg/m^2^ (IQR 4). Ten patients with PC (40%) were (borderline) resectable at baseline. Tissue was available from six patients (resection *n* = 5, biopsy *n* = 1). A median volume of 2.9 mL plasma (IQR 1.0 mL) and 0.7 mL PJ (IQR 0.1 mL, *p* < 0.001) was used for DNA isolation. PJ had a higher median cfDNA concentration (2.6 ng/μL (IQR 4.2)) than plasma (0.29 ng/μL (IQR 0.40)). A total of 41 unique somatic mutations were detected: 24 mutations in plasma (2 *KRAS*, 15 *TP53*, 2 *SMAD4*, 3 *CDKN2A* 1 *CTNNB1*, and 1 *PIK3CA*), 19 in PJ (3 *KRAS*, 15 *TP53*, and 1 *SMAD4*), and 8 in tissue (2 *KRAS*, 2 *CDKN2A*, and 4 *TP53*). The mutation detection rate (and the concordance with tissue) did not differ between plasma and PJ. In conclusion, while the concentration of cfDNA was indeed higher in PJ than in plasma, the mutation detection rate was not different. A few cancer-associated genetic variants were detected in both biomaterials. Further research is needed to increase the detection rate and assess the performance and suitability of plasma and PJ for PC (early) detection.

## 1. Introduction

Pancreatic cancer (PC) is a leading cause of cancer-related death due to asymptomatic progression leading to diagnosis at an unresectable stage in approximately 80% of patients [1,2], while radical surgery of early-stage cancer, or preferably high-grade precursor lesions, is the only chance for long-term survival. So far, extensive imaging-based surveillance programs—following individuals at a high risk of developing PC—have not lived up to their expectations [3]. The use of biomarkers may lead to earlier detection in these programs and, concurrently, support personalization of treatment.

Cancers, including PC, arise as a consequence of accumulating gene mutations. Molecular profiling of tissue collected by fine-needle biopsy (FNB)—to assess these mutations—is less applicable for surveillance purposes, as it is invasive, relies on a visible mass, and is expected to obtain information from a single clone. Additionally, collection during endoscopic ultrasound (EUS) has been challenging due to low tumor cellularity and limited yield of tissue [4,5]. To date, for PC, carbohydrate antigen 19.9 (CA19.9) is the only serum marker implemented in clinical practice. However, for surveillance purposes, CA19.9 is controversial, as elevated values are regularly observed in patients with no or low-grade dysplasia, and low values do not rule out progression. A panel of biomarkers (from different sources) may increase the diagnostic performance.

Gene mutations may be investigated in circulating tumor DNA (ctDNA), which consists of short cell-free DNA fragments released into body fluids by cancer cells due to apoptosis, necrosis, and secretion. Multiple trials have aimed to diagnose PC (and its precursor lesions) based on plasma ctDNA, as it is non-invasively collected and expected to contain information from all present clones [6,7]. However, in patients with PC, ctDNA concentrations in plasma are often below the limits of detection (especially in precursor lesions and early stages) [8,9,10]. Moreover, detected alterations in plasma are not specific for the pancreas, as they may originate from another organ.

Pancreatic juice (PJ) may serve as an alternative biomarker source. A washout from the pancreas can be stimulated by secretin and collected from the duodenal lumen during endoscopy (without pancreatic duct cannulation or tissue sampling). As compared with FNB, PJ does not rely on a visible mass and is expected to contain information on the complete range of tumor clones [11,12]. As compared with blood, biomarkers determined in PJ are likely more pancreas-specific, as this fluid was in continuous close contact with the pancreatic ductal system from which PC originates. In addition, PJ may harbor more ctDNA due to a higher overall cell-free DNA (cfDNA) concentration. On the other hand, it contains enzymes, rendering it important to snap-freeze the sample immediately after collection [12].

We hypothesized that the cfDNA concentration and mutation detection rate by next-generation sequencing (NGS) is higher in PJ than in blood plasma. This study aims to compare the detection rate of gene mutations in (cf) DNA from tissue, plasma, and PJ.

## 2. Results

### 2.1. Patient Cohort

In this study, 26 patients with histologically proven high-grade dysplasia (HGD, *n* = 1) or PC (*n* = 25) were included. These individuals had a median age of 71 years (interquartile range (IQR) 12) and a median BMI of 23 kg/m^2^ (IQR 4), and 7 were women (27%; Table 1). At the time of plasma and PJ collection, diabetes mellitus was present in 10 individuals (39%), and 21 (81%) had symptoms of either biliary obstruction (*n* = 8; 31%; all had a common bile duct (CBD) stent in situ), epigastric pain (*n* = 13; 50%), or weight loss (*n* = 10; 39%). Eight patients had resectable, two had borderline-resectable, and 16 had (locally) advanced diseases. The majority had a CA19.9 level > 37 kU/L (*n* = 20; 77%) and did not undergo chemotherapy before plasma and PJ collection (*n* = 22; 85%). The four individuals who had previously undergone chemotherapy received treatment with either FOLFIRINOX (#18 and #25 for 3 and 8 cycles) or gemcitabine combined with nab-paclitaxel (#16 and #26 for 4 cycles).

Tissue was available from six patients; H&E-stained slides can be found in Appendix A. Fresh-frozen tissue was available and analyzed from five patients (1 HGD: #1; 4 PC: #2–4, #6), and the time between plasma/PJ collection and surgery was 2 months for HGD #1 and 4–6 months for PC #2–4 and #6. FNB specimen was available and analyzed from one patient (#5); this sample was collected on the same day as the PJ/plasma sample.

### 2.2. Cell-Free DNA Concentration and Fragment Length

A median volume of 2.9 mL (IQR 1.0 mL) plasma and 0.7 mL (IQR 0.1 mL, *p* < 0.001) PJ was used for DNA isolation, which resulted in a median yield of 16 ng cfDNA from plasma (IQR 13 ng; median concentration 0.29 ng/μL (IQR 0.40 ng/μL)) and 1630 ng from PJ (IQR 3048 ng, *p* ≤ 0.001; 2.6 ng/μL (IQR 4.2 ng/μL), *p* ≤ 0.001). As compared with plasma, PJ contained more long DNA fragments, as indicated by the long (247 bp) to short-fragment (115 bp) ratio (median 0.26 (IQR 0.07) vs. 0.42 (0.55), respectively, *p* = 0.03; Appendix A). The concentration of cfDNA in plasma was not correlated with that in PJ (*p* = 0.37; Figure 1), yet to the plasma 247/115 ratio (r = −0.61, 95% CI −0.82–−0.23, *p* = 0.003).

The cfDNA concentration (of plasma and PJ) was not associated to age, sex, BMI, previous chemotherapy, location of the solid mass, presence of a CBD stent, and resectability.

### 2.3. Mutation Detection Rate in PJ and Plasma

For NGS, all isolated cfDNA (up to 25 ng) from plasma samples (range 4.7–25 ng) and 25 ng from PJ was used. For the Oncomine panel (#1–4), the number of reads ranged from 4.8 million to 7.0 million. For the Swift panel, the median number of used sequence reads was 2.8 million (IQR 1.4) for plasma and 2.6 million (IQR 0.9 million) for PJ (*p* = 0.23). For the number of reads per panel and coverage, see Appendix A.

A total of 41 unique somatic mutations were detected (Figure 2); 24 in plasma (2 *KRAS*, 15 *TP53*, 2 *SMAD4*, 3 *CDKN2A*, 1 *CTNNB1*, and 1 *PIK3CA*), 19 in PJ (3 *KRAS*, 15 *TP53*, and 1 *SMAD4*), and 8 in tissue (2 *KRAS*, 2 *CDKN2A*, and 4 *TP53*). At least one mutation was present in 17 (65%) plasma samples, 16 (61%) PJ samples, and 5 (83%) tissue samples (*p* = 0.22). The median number of mutations was 2 (IQR 1) for plasma, 1 for PJ (IQR 1), and 2 for tissue (IQR 2; *p* = 0.74). The number of mutations per sample was not associated with the input cfDNA concentration, number of reads, Alu247/Alu115 ratio, age, sex, BMI, concentration of CA19.9, presence of DM, previous chemotherapy, resectability, CBD-stent in situ or the location of the solid mass (*p* > 0.05; Figure 2).

*KRAS* mutations were present in 13 of 26 patients (50%): 5 plasma samples (19%), 9 PJ samples (35%), and 4 tissue samples (67%; Figure 3). The detection rate in tissue was higher than that in plasma (*p* = 0.02); no difference in the detection rate was found between PJ and plasma. As compared with tissue, plasma was concordant in one patient (#6, *KRAS* p.G12D); PJ was concordant in two patients (#4 and # 6, *KRAS* p.G12D). The concordance and detection rate were not related to resectability or the previous administration of chemotherapy. Concordance of *KRAS* mutations between PJ and plasma was seen in three patients (#6, #17: G12D, #16 G12V; Figure 2).

*TP53* mutations were most prevalent and detected in 19 of 26 patients (73%): in 12 plasma samples (46%), 13 PJ samples (50%), and 4 tissue samples (67%; *p* > 0.05). For *TP53*, no concordance between tissue and plasma was seen, yet in two patients, concordance between PJ and plasma was seen (#13: R273H and #20: R248Q). The *TP53* P72R homozygous germline variant (R72/R72) was present in 11 patients (5 (45%) had ≥1 *TP53* mutation), and the *TP53* P72R heterozygous germline variant (P72/R72) was present in 12 patients (12 (100%) had ≥1 *TP53* mutation). Three patients had no P72R variant (P72/P72; 2 (67%) had ≥1 *TP53* mutation). As expected for a germline variant, the concordance between plasma and PJ was 100%.

## 3. Discussion

This study compares the detection rate of DNA mutations in PJ and plasma, which were (unexpectedly) found to be similar. While the concordance of the TP53 P72 variants was 100%, the concordance between biomaterials for variants with a lower allelic frequency was small. Additionally, both the total cfDNA concentration and the Alu247/Alu115 were higher in PJ than in plasma.

cfDNA analyses in plasma and secretin-stimulated PJ can serve as non-invasive method for repetitive (longitudinal) sampling in a surveillance program. As (cancer) cells shed information to surrounding tissues and body fluids, it is expected that biomarkers are able to detect PC at an earlier stage. Additionally, in case of a suspicion of PC, obtaining a pancreatic biopsy is often challenging and requires multiple sampling efforts. Currently available biomarkers (such as CA19.9) have a high false-positive rate and are therefore not able to fulfill this need [13]. Additionally, as an increasing number of targeted therapies are being developed, cfDNA analysis may be able to identify cancers eligible for personalized approaches (such as HER2 or BRAF targeted therapy or platinum-based chemotherapy for *BRCA2* mutated PC).

While several studies have described ctDNA analyses in plasma or PJ for PC detection, we are the first to compare these two sources head to head. We hypothesized the superiority of PJ for the detection of PC (based on a higher concentration of DNA). Indeed, the current study showed a significantly higher cfDNA concentration as well as Alu247/Alu115 ratio in PJ than in plasma. This means that the cfDNA in PJ may potentially be contaminated with high concentrations of genomic DNA (as a result of cell decay during collection). Conversely, the higher Alu247/Alu115 ratio in PJ may be due to its close relation with PC and may be a measure of cell necrosis (rather than apoptosis), as necrosis produces longer cfDNA fragments [14,15,16,17]. If so, the Alu247/Alu115 ratio in PJ may be a measure of tumor volume and response to therapy. Furthermore, smaller lesions may be detectable earlier in PJ than in plasma. However, further research is needed to verify this.

For *KRAS*, our analysis showed a low *KRAS* detection rate in PJ (35%, as compared with 60% reported in the literature) [18] as well as serum. In the current cohort, none of the patients had metastatic disease and therefore may have had insufficient advanced disease to generate an allelic frequency above the limit of detection (while potentially present as driver mutation in the tumor). Unfortunately, this would limit the use of this analysis for detection of lesions prior to them becoming visible on imaging. We do not believe that this has been caused by the use of secretin-stimulated PJ (as compared with the ‘purer’ PJ collected by ampullary cannulation) because other studies showed a similar *KRAS* detection rate when using secretin-stimulated PJ or pure PJ (collected by ampullary cannulation). However, no head-to-head comparison was performed in any of these studies.

For *TP53*, this study shows a detection rate of 50% in PJ, which is higher than the detection rate in PJ previously described (43%) [13]. This may have been caused by our strict selection criteria to call a mutation. For plasma samples, the *TP53* detection rate was 46%. In patients >65 years (23 of 26 patients in this cohort), there is a potential for confounding plasma *TP53* variants due to clonal hematopoiesis, but the low allelic frequency makes this unlikely [19]. Our data imply that the prevalence of somatic *TP53* mutations is higher in patients with a germline heterozygous *TP53* P72R (P72/R72) variant than in those with a homozygous variant (R72/R72). This is consistent with the study of De Souza et al. (2021) [20,21], who showed enrichment of *TP53* genes in ovarian cancer cells harboring the P72 variant. Furthermore, van der Sijde et al. (2021) [22] showed that a P72R homozygous variant was associated with early tumor progression and poor overall survival. More research is required to evaluate the role of these P72 variants in the development and prognosis of PC.

This study has several limitations. Tissue (either resection or FNB specimen) was only available in six patients (for logistic reasons). Thus, we were unable to evaluate which of the observed mutations (in plasma or PJ) are actually reflective of tumor tissue. A few patients underwent previous chemotherapy, which may have caused a lower detection rate for these samples. To evaluate the performance of a test, a case-control design would be more valuable than the current design comparing detection rates. Another limitation of this study is the use of different kits for different samples. Based on our data, it is unclear which of the used kits performs better. At last, sample handling was different between plasma and PJ samples. Plasma was stabilized in EDTA anticoagulant and a cell preservative (CellSave)—proven to preserve the cfDNA quality and somatic variant detection ability in the serum for at least 96 h [23]—and PJ was snap frozen. This may have caused the difference in results between plasma and PJ.

In conclusion, this head-to-head comparison of plasma and PJ confirms a higher DNA concentration and Alu247/Alu115 ratio in PJ, yet it does not show the superiority of PJ over plasma in the detection rate of PC-related mutations. The exploratory nature of these results demands further research on sample handling (such as the used volumes, the use of cell preservation tubes to reduce genomic DNA for PJ, and the possibilities of scavenging background noise or the role of DNA sequencing kits) to increase the mutation rate in these samples and evaluate the potential applicability of PJ for molecular analysis. Furthermore, research with a case-control design is needed to compare the accuracy of mutations in PJ and plasma.

## 4. Materials and Methods

### 4.1. Study Design and Patient Inclusion

Plasma, PJ, and tissue samples were prospectively collected at the Erasmus University Medical Center in Rotterdam as part of two clinical studies: 1. KRASPanc study (MEC-2018-038), concerning patients with (suspected) sporadic PC undergoing diagnostic EUS or fiducial placement for stereotactic radiotherapy; and 2. PACYFIC study (MEC-2014-021), involving individuals undergoing surveillance for suspected neoplastic pancreatic cysts. See Appendix A for a description of the KRASPanc and PACYFIC study cohorts.

Participants of these clinical studies were included for the present study based on the presence of (pathology-proven) HGD or PC and the availability of both plasma and PJ samples. Patients who had undergone radiotherapy or pancreatic resection prior to PJ collection were excluded. The availability of tissue samples was not an inclusion or exclusion criterion, and selection of tissue for NGS was based upon the availability of fresh-frozen tissue (from FNB or tissue) that had been performed during clinical workup. No formal sample size analysis was performed due to the explorative nature of this study.

The institutional medical center ethical review board approved the studies, and included individuals gave written informed consent before enrolment. The studies were carried out according to the ethical principles for medical research involving human subjects from the World Medical Association Declaration of Helsinki.

### 4.2. Biomaterial Collection

PJ collection was performed with a linear echoendoscope (Pentax Medical, Tokyo, Japan) by experienced endo-sonographers. After insertion of the tip of the echoendoscope into the D2 segment of the duodenum, secretion of PJ was stimulated by intravenous injection of human secretin (16 µg/patient, ChiRhoClin, Burtonsville, MD, USA). Suction through the endoscopic channel was applied immediately after secretin injection for eight minutes by positioning of the tip close to the ampullary orifice [12]. Within 10 min after collection, juice was aliquoted and snap frozen. Samples were stored at −80 °C until use.

Plasma samples were collected by venipuncture in CellSave tubes (CellSearch, Bryn Athyn, PA, USA, #7900005) on the same day as the PJ collection. After collection, centrifugation at 1600 rpm for 10 min was performed, and samples were aliquoted and stored at −80 °C until use. Before cfDNA isolation, samples were centrifuged at 16,000 rpm for 10 min, and DNA was isolated from the supernatant. Tissue samples were freshly frozen after resection (patient #1–4, #6) or FNB (#5) at the same day. Routine hematoxylin and eosin (H&E) stained sections from tissue samples were assessed for tumor cellularity by a pathologist, and areas enriched for tumor cells were identified and manually micro-dissected.

### 4.3. DNA Isolation

For cfDNA isolation, an automatic bead-based Maxwell RSC cfDNA Plasma kit (Promega, Fitchburg, WI, USA, AX1115) was used according to the manufacturer’s instructions. The choice of a kit was based on previous results by our group [12]. To quantify the concentration of total (double-stranded) DNA, a Quant-iT dsDNA High-Sensitivity Assay Kit (Thermo Fisher Scientific, Waltham, MA, USA) was employed according to the manufacturer’s instructions. For mutational cfDNA analysis of plasma and PJ samples, DNA libraries were prepared using 25 ng cfDNA input from PJ. From plasma samples, all available cfDNA (up to 25 ng) was used.

To prepare the tumor tissue samples for sequencing, the tissue was washed twice with PBS and subsequently treated with 550 µL lysis buffer and 20 µL protease overnight at 37–55 °C. RNase A (3 µL) was added and incubated for 15–60 min at 37 °C. Subsequently, a Chemagic MSM1 isolation robot (PerkinElmer Chemagen Technology, Baesweiler, Germany) was used to isolate the DNA according to the manufacturer’s recommendations.

### 4.4. Deep Sequencing and Data Analysis

First, we performed a pilot and solely evaluated the plasma and PJ samples of four patients (#1–4) using the Oncomine Colon cfDNA Assay (Thermo Fisher Scientific, Waltham, MA, USA, Appendix A), with library preparation according to the manufacturer’s recommendations. Results were contrasted against sequencing data derived for clinical purposes from tissue (patients #1–6) using an in-house-generated pan-cancer AmpliSeq (Thermo Fisher Scientific, Waltham, MA, USA) panel covering 100% of *CDKN2A*, *KEAP1*, *PTEN*, *STK11*, *TP53*, and hotspots in other cancer genes (*BRAF*, *CTNNB1*, *EGFR*, *ERBB3*, *FBWX7*, *GNAS*, *KRAS*, *PIK3CA*, *SF3B1*, and *SMAD4*, among others; Appendix A). Again, libraries were prepared according to the manufacturer’s recommendations. Subsequently, for all three biomaterials, template preparation was performed using an Ion Chef system, and sequencing was performed using Ion GeneStudio S5 Prime System on 540 chips with an Ion 540 Chef Kit. Data were analyzed with Variant Caller v5.10.0.18 and annotated using ANNOVAR [24].

Secondly, we performed sequencing on an extended panel of genes to evaluate the presence of additional mutations in a larger group. For plasma and PJ samples of patients #5–26, cfDNA concentrations were first measured by real-time qPCR using Alu115 primer pairs (forward: 5′-CCTGAGGTCAGGAGTTCGAG-3′ and reverse: 5′-CCCGAGTAGCTGGGATTACA-3′) using initial denaturation (at 95 °C for three minutes—at 95 °C for five seconds—at 62 °C for 30 s; Swift Biosciences, Ann Arbor, MI, USA). Additionally, real-time qPCR was performed using Alu247 primers (forward: 5′-GTGGCTCACGCCTGTAATC-3′ and reverse: 5′-CAGGCTGGAGTGCAGTGG-3′) using the same PCR program as for Alu115. As cfDNA exhibits a narrow size distribution of ±167 bp, Alu115-qPCR results quantify the total amount of cfDNA, while the Alu247/Alu115 ratio illustrates the cfDNA integrity. DNA libraries were prepared by multiplex PCR using the Accel-Amplicon 57G Plus pan-cancer profiling panel (Swift Biosciences, Ann Arbor, MI, USA), which covers 286 amplicons of 57 genes, amplified for 25 cycles in total (as per protocol), followed by the ligation of Illumina adaptors with sample specific indices. See Appendix A for the similarities and differences between used panels. Indexed sequencing libraries of PJ, plasma, and tissue cfDNA were pooled, and 250 paired end or 300 base pair were sequenced on two flow cells of an Illumina MiSeq sequencer (Illumina, San Diego, CA, USA). From the reads, adapter sequences were trimmed and aligned to the human GRCh38 reference genome using BWA mem. Variant calling was performed using BCFtools combined with in-house scripts.

Non-synonymous variants were considered as true mutations if these had a variant allele frequency (AF) > 0.1%, ≥5 mutated reads, and if they are ≥5 times described as somatic mutation in PC patients in the databases of COSMIC, OncoKB, and http://www.cancerhotspots.org (accessed on January 2023). Mutations with allele frequencies higher than 49% were considered as homozygous or heterozygous germline mutations, not ctDNA mutations, and were excluded. The prevalence of *TP53* P72R homozygosity and heterozygosity was noted, as it may modify the effect of *TP53* hotspots mutants (whether it is associated with an increased risk of cancer remains highly controversial) [20,21,25,26]. The potential pathogenicity of somatic mutations was evaluated using ClinVar (https://www.ncbi.nlm.nih.gov/clinvar/) and COSMIC (http://grch38-cancer.sanger.ac.uk/cosmic). Variants judged as benign were not tabulated. Recurrent mutations occurring in ≥10 samples were compared with prevalence in PC in the COSMIC database in case of an unusual high (100×) prevalence; PCR errors (e.g., polymerase slippage) and sequencing errors (e.g., substitution errors and pseudogenes) were considered. These variants were only considered in case of AF > 0.5%. For instance, this applied to *TP53* p.G245D, which was present in 25 of the 34 samples (74%) analyzed with the Accel-Amplicon 57G Plus pan-cancer profiling panel, yet present in only 9/2829 (0.3%) investigated patients in the COSMIC database. This also applied to *SMAD4* p.R135* present in 17/34 (50%) samples (COSMIC: 7/2829) and *PIK3CA* p.E545A in 10/34 (29%) samples (COSMIC: 4/2273).

### 4.5. Statistical Analysis

All statistical analyses were performed with SPSS (Statistical Package for the Social Sciences, version 27, SPSS Inc., Chicago, IL, USA); figures were generated using GraphPad (GraphPad Prism version 9, GraphPad Software, La Jolla, CA, USA). *p*-values < 0.05 were considered significant.

Descriptive data were expressed as medians with IQR or percentages. Further statistical analyses were performed using a Wilcoxon paired-samples test or Spearman’s rank-order correlation for continuous variables.

## Figures and Tables

**Figure 1 ijms-24-13116-f001:**
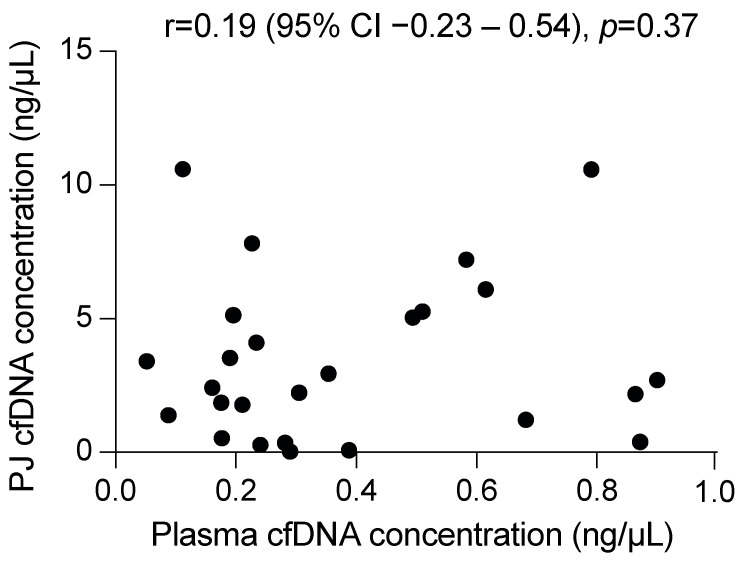
The DNA concentration in pancreatic juice (PJ) is not correlated with that in plasma. Displayed correlation coefficient is a Pearson correlation.

**Figure 2 ijms-24-13116-f002:**
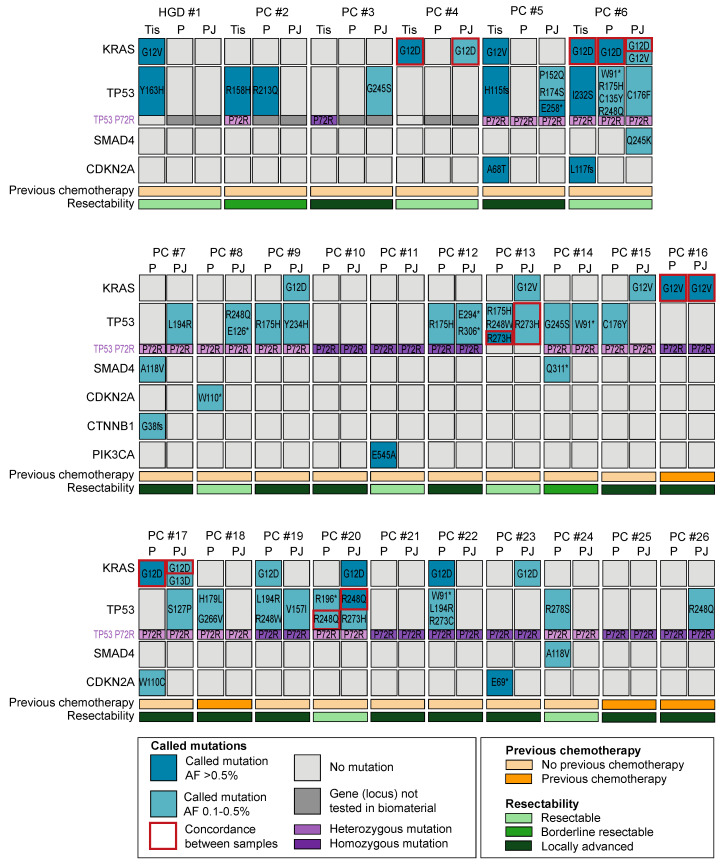
Schematic overview of gene mutations in plasma (P) and pancreatic juice (PJ) for 1 patient with high-grade dysplasia (HGD; #1) and 25 patients with pancreatic cancer (PC; #2–#26). Only genes showing an alteration are presented. Gene mutations in plasma and PJ #1–4 were evaluated using the Oncomine colon cfDNA assay, and for #5–26, the Accel-Amplicon 57G Plus pan-cancer profiling panel was used. For the tissue (#1–4, #6) and biopsy samples (#5), an in-house-generated pan-cancer panel was used. Concordance of *KRAS* between PJ and plasma was seen in three patients (#6, #16, and #17), whereas concordance of *TP53* between PJ and plasma was seen in two patients (#13 and #20). Furthermore, patients with a P72R heterozygous variant had a higher TP53 detection rate than those with a homozygous variant. HGD = high-grade dysplasia, PC = pancreatic cancer, Tis = tissue, P = plasma, PJ = pancreatic juice, and AF = allelic frequency. * indicates a nonsense mutation.

**Figure 3 ijms-24-13116-f003:**
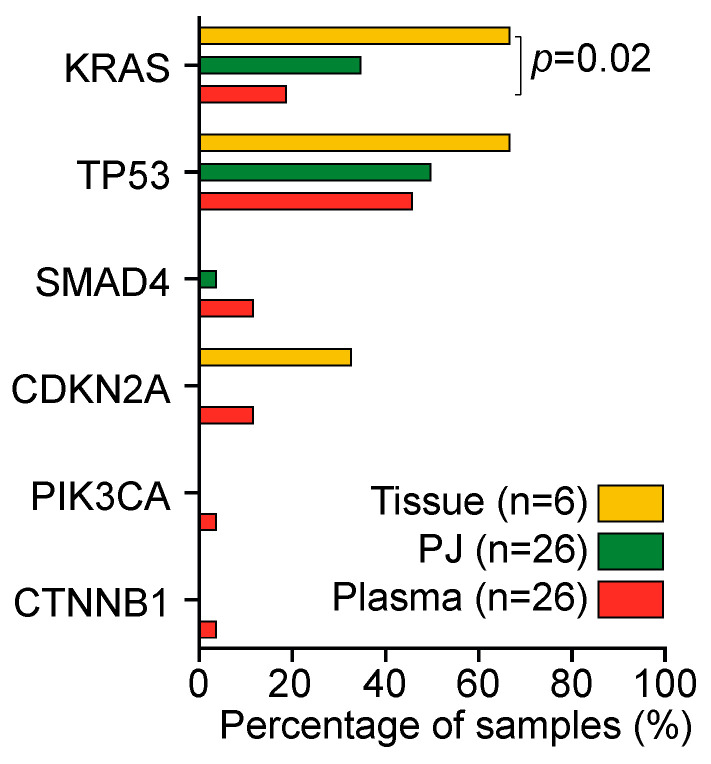
Detection rate of mutations in tissue (yellow), pancreatic juice (PJ; green), and plasma (orange). KRAS mutations were detected significantly more often in tissue as compared with plasma; KRAS detection rate between PJ and plasma was not different. Percentages were compared with a Kruskal–Wallis test.

**Table 1 ijms-24-13116-t001:** Patient characteristics (all inclusions).

	Total Cohort (*n* = 26)
Age, median (IQR)	71 (12)
Sex, *n* women (%)	7 (27)
BMI ^¶^, median in kg/m^2^ (IQR)	23 (4)
Smoking, *n* (%)	
No	6 (23)
Currently	7 (31)
Former (>2 years ago)	11 (42)
Unknown	1 (4)
Diabetes mellitus, *n* present (%)	10 (39)
Any symptom, *n* (%)	21 (81)
Jaundice	8 (31)
Epigastric pain	13 (50)
Weight loss	10 (39)
CA19.9 > 37 kU/L, *n* (%)	20 (77)
Treatment naive, *n* (%)	21 (81)
Resectability of PC, *n* (%)	
Resectable	8 (31)
Borderline resectable	2 (8)
Locally advanced PC	16 (62)
Location mass, *n* (%)	
Uncinate/head	16 (62)
Neck/corpus	7 (27)
Tail	3 (12)
CBD stent in situ, *n* (%)	8 (31)

^¶^ BMI is missing for 5 patients. IQR = interquartile range, PC = pancreatic cancer, CBD = common bile duct.

## Data Availability

This study was conducted as part of the KRASPanc study (MEC-2018-038) and PACYFIC study (MEC-2014-021; 1 sample).

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
