# Peer review of "Mutation Analysis of Pancreatic Juice and Plasma for the Detection of Pancreatic Cancer"

_ijms, 2023, doi:10.3390/ijms241713116_

Round 1

Reviewer 1 Report

This is a study analysing recovery of tumor specific mutations in cfDNA isolated from PJ as source compared to plasma in 25 cases of histology proven PC and one patient with IPMN with HGD. The authors claim that results demonstrate that PJ as source is not superior to plasma.

The number of cases included in this study is low (n=26), thus the conclusions that can be drawn are limited. Tissue analysis is present only in six cases, and three different NGS analysis platforms were used. 

The authors should state what would be their take-home for this trial. The conclusions seem to be very preliminary. What would be the next steps to take?

Given these limitations, I have the following further concerns:

-       What was the median amount of DNA input used for NGS in plasma (range was 4,7-25ng)? Can the authors exclude that the lower amount of cfDNA input from plasma might have biased the result?

-       The authors should comment on processing time of P versus PJ samples. Are CellSave tubes validated for isolation of cfDNA?

-       Did the concentration of cfDNA in PJ correlate to CRP (if available)?

-       What was the mean coverage of NGS runs (Fig. S3 not included in suppl. Files)?

-       Can the authors speculate about the relatively low concordance between P and PJ especially for TP53? 

-       Did the authors have the opportunity to sequence more than one tumor region in the tissue block (tumor heterogeneity)?

none

Reviewer 2 Report

This manuscript focuses on a pilot study pointing on the identiifcation of a novel promising biological source (pancreatic juice) in the diagnosis of pancreatic lesions on a retrospective series of cases. The manuscript highlights a concise and well documented methodological and results section in accordance with a fascinating topic. In my opinion, minor considerations should be approached to improve the availability of the manuscript on this journal

- Introduction section. The authors clearly identify the technical need behind this project. In my opinion, they should improve this section stressing the relevance of integrating biological source for cfDNA analysis. Morover, a brief description of technical approaches available for this purpose may represent a plus for this manuscript

- Methodological section appears well documented. In this scenario, I would reccomend discussing the role of tissue sampels as referral material in order to overcome this limitation.

- Results section shows interesting molecular data in line with previously cited methodological aspects. In this setting, please, could the authors focus on mutant allele fraction? Could this paramter may be object of additional analysis improving result section?

- In the discussion section, please, could the authors underline the main limitations of the study?

Minor english editing

Reviewer 3 Report

The article entitled “Mutation analysis of pancreatic juice and plasma for the detection of pancreatic cancer” by Levink et al., provides the evidence that pancreatic juice is as representative as plasma sample to diagnose pancreatic cancer patients according to their main key mutations. Furthermore, paired samples demonstrate that pancreatic juice has the same mutations that plasma samples to detect cfDNA. However, the lack of tumour tissues  unables de identification of higher rates of TP53, and the match with PJ or plasma mutations. The manuscript is overall well written, introduced and discussed and the conclusions are supported by results.

Please find below my points to improve the quality of the manuscript:

1)Please review the writing style of mutations, according to international consensus DNA genes they must be written in upper case and italics, proteins in upper case and regular and mRNAs in lower case and regular.

2)Please replace the word “create” since only God can create, use better “generate”.

3)Statistical analysis denotes that only has been used p-values to considered significancy. Do you have included in the analysis FDR? If not justify.

4)Wilcoxon test is used for non-parametric variables and Pearson is used for correlations between parametric variables. Please justify with K-S test the normality of the variables studied. If N is not enough to get a parametric distribution recalculate with Spearman test.

5)In result section, rephrase the subtitles as a result, they seem materials and methods.

Is good

Round 2

Reviewer 1 Report

The authors have addressed the further concerns raised by this reviewer. The preliminary nature of this report however remains. The authors show that cfDNA analysis in PJ is feasible however the conclusion that results demonstrate that PJ as source is not superior to plasma can not be convincingly derived from the data presented. 

none

Author Response

The sole remaining criticism (of reviewer 1) is that the study provides preliminary data. We fully acknowledge that our study is of an exploratory nature, which we can further emphasize in the manuscript. Pancreatic juice evaluation was hampered by a paucity of human secretin, which was only recently relieved. Thus, now is the time for exploratory research. As a first step we evaluated if the higher DNA concentration in pancreatic juice – as compared to plasma – also leads to a better mutation detection rate. This study did not show superiority of pancreatic juice, however it underlines other differences (and similarities) between these biomaterials, as well as highlights those factors during collection and analysis that require further development and investigation. For instance, comparison of sample handling may increase the detected mutation rates. Potential ways to improve the mutation detection rate could be to use cell preservation tubes to reduce genomic DNA within pancreatic juice and scavenging background noise during analyses. This provides new inspiration for further research.
Exploring this new biomarker source is of vital importance, as early detection of pancreatic cancer in individuals undergoing imaging-based surveillance has been challenging thus far. The detection of a biomarker signature in pancreatic juice may stratify the risk of developing pancreatic cancer and bring forth a surveillance program with tailored intervals and modalities. More specifically, pancreatic juice is a promising biomarker source, as it constitutes a wash-out of the pancreatic ductal system and has been in close contact with the ductal cells from which pancreatic cancer originates.

Reviewer 3 Report

Many thanks for provide an amended version.

it is Ok

Author Response

Thank you